# A Multicenter Study Evaluating the Discontinuation of Eculizumab Therapy in Children with Atypical Hemolytic Uremic Syndrome

**DOI:** 10.3390/children9111734

**Published:** 2022-11-11

**Authors:** Saeed AlZabali, Sawsan AlBatati, Khawla Rahim, Hassan Faqeehi, Abubaker Osman, Abdulaziz Bamhraz, Mohammed A. Saleh, Jameela A. Kari, Majed Aloufi, Loai Eid, Haydar Nasser, Abubakr Imam, Entesar AlHammadi, Omar Alkandari, Mohammed Al Riyami, Sidharth Sethi, Christoph Licht, Khalid A. Alhasan, Abdulkarim AlAnazi

**Affiliations:** 1Pediatric Nephrology Department, King Fahad Medical City, Riyadh 11525, Saudi Arabia; 2Division of Pediatric Nephrology, Children’s Hospital-McMaster University, Hamilton, ON L8N 3Z5, Canada; 3Section of Medical Genetics, Children Hospital, King Fahad Medical City, Riyadh 12231, Saudi Arabia; 4Pediatric Nephrology Unit, Faculty of Medicine and Pediatric Nephrology Center of Excellence, King Abdulaziz University Hospital, King Abdulaziz University, Jeddah 21589, Saudi Arabia; 5Pediatric Nephrology and Pediatric Kidney Transplantation, Prince Sultan Military Medical City, Riyadh 12231, Saudi Arabia; 6Pediatric Nephrology Department, Dubai Hospital, Dubai 14660, United Arab Emirates; 7Pediatric Nephrology Department, King Fahad Military Hospital, Jeddah 21589, Saudi Arabia; 8Pediatric Nephrology and Hypertension, Sidra Medicine, Doha P.O. Box 26999, Qatar; 9Pediatrics Nephrology Department, Dubai Health Authority, Dubai 14660, United Arab Emirates; 10Division of Pediatric Nephrology, Mubarak Al-Kabeer Hospital, Street 103, Jabriya 46300, Kuwait; 11Pediatric Nephrology Department, The Royal Hospital, P.O. Box 1331, Muscat 111, Oman; 12Pediatric Nephrology, Kidney Institute, Medanta, The Medicity Hospital, Gurgaon 122007, Haryana, India; 13Division of Nephrology, The Hospital for Sick Children, University of Toronto, Toronto, ON M5G 1X8, Canada; 14Pediatric Department, College of Medicine, King Saud University, Riyadh 12231, Saudi Arabia; 15Pediatric Kidney Transplant Section, Organ Transplant Center, King Faisal Specialist Hospital and Research Center, Riyadh 12231, Saudi Arabia

**Keywords:** atypical hemolytic uremic syndrome, discontinuation of eculizumab, thrombotic, microangiopathy, HUS relapse

## Abstract

Background: Atypical hemolytic uremic syndrome (aHUS) is a rare, life-threatening thrombotic microangiopathy (TMA), which has been treated successfully with eculizumab. The optimal duration of eculizumab in treating patients with aHUS remains poorly defined. Methods: We conducted a multicenter retrospective study in the Arabian Gulf region for children of less than 18 years of age who were diagnosed with aHUS and who discontinued eculizumab between June 2013 and June 2021 to assess the rate and risk factors of aHUS recurrence. Results: We analyzed 28 patients with a clinical diagnosis of aHUS who had discontinued eculizumab. The most common reason for the discontinuation of eculizumab was renal and hematological remission (71.4%), followed by negative genetic testing (28.6%). During a median follow-up period of 24 months after discontinuation, 8 patients (28.5%) experienced HUS relapse. The risk factors of recurrence were positive genetic mutations (*p* = 0.020). On the other hand, there was no significant relationship between the relapse and age of presentation, the need for acute dialysis, the duration of eculizumab therapy before discontinuation, or the timing of eculizumab after the presentation. Regarding the renal outcomes after discontinuation, 23 patients were in remission with normal renal function, while 4 patients had chronic kidney disease (CKD) (three of them had pre-existing chronic kidney disease (CKD) before discontinuation, and one case developed a new CKD after discontinuation) and one patient underwent transplantation. Conclusions: The discontinuation of eculizumab in patients with aHUS is not without risk; it can result in HUS recurrence. Eculizumab discontinuation can be performed with close monitoring of the patients. It is essential to assess risk the factors for relapse before eculizumab discontinuation, in particular in children with a positive complement variant and any degree of residual CKD, as HUS relapse may lead to additional loss of kidney function. Resuming eculizumab promptly after relapse is effective in most patients.

## 1. Introduction

Atypical hemolytic uremic syndrome (aHUS) is a rare, life-threatening form of TMA that is caused by uncontrolled complement activation in the alternative pathway (AP) [1,2]. It is characterized by microangiopathic hemolytic anemia (MAHA), thrombocytopenia, and evidence of endothelial cell destruction, which result in ischemic end-organ damage [1,3,4]. Genetic defects in complement-related factors or acquired autoantibodies to complement factors have been found in 50 to 60% of patients [2]. The initial therapy for aHUS is supportive, focusing on managing acute kidney injury and systemic complications. Plasma exchange and eculizumab, a complement inhibitor, offer specific forms of treatment [5]. Eculizumab, the first approved treatment for patients with aHUS, is a humanized monoclonal complement inhibitor that binds to C5, preventing its cleavage to C5a and C5b and the subsequent activation of the complement terminal pathway (TP) and membrane attack complex (MAC) formation [6,7]. Eculizumab has profoundly changed the outcomes for patients with aHUS compared with other traditional options, such as plasma therapy [7]. The necessary duration of eculizumab treatment in aHUS is unknown [8]. The European Commission’s Summary of Product Characteristics and the Food and Drug Administration’s Full Prescribing Information, both released in 2014, do not take a stance on the treatment duration [9]. The reasons for discontinuing eculizumab therapy reported in the literature are the protection of patients from the risk of the potentially devastating side effects of meningococcal infection, the requirement for repeated infusions, and the high cost of treatment [1,10]. There is a substantial risk of relapse when stopping eculizumab in patients with aHUS. Selected groups of patients appear to discontinue eculizumab and remain relapse-free for several years [11]. This study aims to report the rate and risk factors for the relapse and renal outcome of the patients after discontinuation

## 2. Materials and Methods

A retrospective multicenter study for the period from June 2013 to June 2021 was carried out across five countries in the Arabian Gulf region, in the Kingdom of Saudi Arabia (5 centers) and the United Arab Emirates (2 centers), and in one center in each of Oman, Qatar, and Kuwait. Patients were included in the study if they were under 18 years of age, were diagnosed with aHUS (first episode or relapse), received at least two months of eculizumab treatment before discontinuation, and had at least six months of follow-up after the initial eculizumab discontinuation unless HUS recurred sooner. Patients on chronic dialysis (defined as three months’ of dialysis) at the time of discontinuation and patients with other causes of thrombotic microangiopathy, including infection with Shiga-toxin-producing bacteria; a disintegrin and metalloproteinase with a thrombospondin type 1 motif, member 13 (ADAMTS13) deficiency; and conditions associated with secondary HUS (malignancy, malignant hypertension, the use of drugs, autoimmune diseases, infections, cobalamin C deficiency, transplantation, and pregnancy) were excluded. Primary aHUS was defined [10] by at least two of the following criteria: (i) thrombocytopenia (platelet count < 150,000 9/L); (ii) mechanical hemolytic anemia (Hb < 10 g/dL, lactate dehydrogenase (LDH) serum level > upper limit of normal, low haptoglobin, and the presence of schistocytes on blood smear); (iii) acute kidney injury (serum creatinine upper limit of normal for age or increased >25% compared to baseline). 

The diagnosis of relapse was based on the criteria of the initial diagnosis of aHUS. Remission (at the time of eculizumab discontinuation) was defined as lactate dehydrogenase (LDH) < 1.5 upper limit of normal, the absence of hemolysis (hemoglobin started to increase or returned to baseline), platelet count > 150,000 9/L, and serum creatinine decreased by 25% compared with baseline. The data collected were on the demographic information, clinical history, laboratory investigations (hemoglobin, platelets, reticulocytes count, blood smear, LDH, haptoglobin, serum creatinine, C3, C4, complement factor H, I, B, CH50/AHA50 level, blood culture, urine culture, urine for protein/creatinine ratio), renal biopsy findings, genetic studies, the timing of eculizumab (after presentation), requirement of acute dialysis before eculizumab therapy, the duration of eculizumab before discontinuation, reason for the discontinuation of eculizumab therapy, follow-up time after discontinuation, laboratory results at the time of the discontinuation of eculizumab and thereafter, estimated glomerular filtration rate (eGFR) at the last visit after discontinuation, relapse after discontinuation, possible trigger for new HUS recurrence episode, timing of starting eculizumab after relapse, trial of discontinuation after treatment relapse episode, and outcome after discontinuation. The CKD was defined based on the KDIGO definition 2012.

The data were saved in a password-protected Excel folder, with only the investigators having access to the data. The patient information remained confidential, with all identifiers removed from the stored data. IRB approval was obtained from all contributing centers.

### Statistical Analysis

All data analyses were performed using Statistical Packages for Software Sciences (SPSS) version 21 (Armonk, New York, NY, USA, IBM Corporation). The descriptive statistics are presented using counts and proportions (%). A *p*-value cutoff point of 0.05 at 95% CI was used to determine the statistical significance. The research team performed the statistical association and calculation process; special software may have been used during the analysis. To compare patients with vs. without relapse, continuous variables were tested using the Wilcoxon signed-rank test, and categorical variables were tested using Fisher’s exact test. A Kaplan–Meier analysis of relapse-free survival was performed

## 3. Results

### 3.1. Patient Cohort

Twenty-eight patients were included in this study. The mean age of the patients was 3.47 years (SD ± 2.96), with the majority being males (70.4%). Eleven patients (39.2%) had a positive family history of the disease, while twenty (71.4%) patients had positive consanguinity. The mean values for the presentation of hemoglobin, platelets, and LDH were 8.1 g/dL, 84.5 10^9^/L, and 2106 U/L, respectively. Almost 50% of patients received eculizumab during the first four days of presentation, while 35.7% of patients received eculizumab after four days. Acute dialysis was required in 64.3% of the children before the initiation of eculizumab. After receiving eculizumab, 82.1% (*n* = 23) showed complete remission. The median duration of eculizumab treatment before discontinuation was 12 months. Seven (25%) patients had received eculizumab therapy for 12 to 24 months, while 6 (21.4%) patients had received eculizumab for 24–48 months and the same percentage had received eculizumab for >3–6 months. The remaining patients had variable durations (Table 1). 

The most common reason for the discontinuation of eculizumab therapy was that the patient was in remission (71.4%), followed by negative genetic testing (28.6%). Other reasons included family preference (10.7%), DGKE mutation (10.7%), and loss to follow-up (7.4%). The median follow-up after discontinuation was 24 months.

### 3.2. Gene Mutation and Relapse

Of the 28 patients, sixteen (57.2%) showed positive genetic testing results. Eleven (39.2%) patients showed no genetic mutation, and one (3.6%) had no genetic testing done (Table 2). 

CFHR1 and CFHR3 mutations were reported in 5 patients (17.8%). Only one of them relapsed after 12 months of discontinuation. Three patients (10.7%) had DGKE mutations, and one of them relapsed after 12 months of discontinuation. Two patients (7.1%) had combined mutations, one of them had combined MCP/CD46 and CFHR1, CFHR3 mutations, the other patient had CFI and CFB mutations; both relapsed after 3 months of discontinuation. MCP/CD46 mutations were reported in two patients (7.1%); one experienced a relapse after 15 months of eculizumab discontinuation. CFH mutations were reported in two patients (7.1%); one had a relapse after 2 months of discontinuation. The only patient with a C3 mutation (3.5%) relapsed after 3 months of discontinuation. Eleven patients (39.2%) had no detected mutations; one of them had a relapse after 2 months of discontinuation. One patient underwent no genetic testing without a history of relapse (Figure 1)

### 3.3. Relapse of aHUS

Eight (28.5%) patients experienced HUS relapse after the discontinuation of eculizumab. Seven of them (25%) had a detected gene mutation. During a median follow-up period of 24 months after discontinuation, most of the patients (75%) restarted eculizumab within 24–48 h. The median time to HUS relapse after discontinuation was 3 months. We observed that 5 patients experienced a relapse within 1 to 3 months, with upper respiratory tract infection (URTI) being the most common triggering factor. None of the patients required dialysis at the time of relapse. All of the patient’s hematological and renal functions returned to pre-cessation values following the resumption of eculizumab after HUS relapse. The prevalence of no relapse was significantly more common among those with no mutation (*p* = 0.020). On the other hand, there was no significant relationship between relapse and the need for acute dialysis (*p* value 0.669), the duration of eculizumab therapy before discontinuation (*p* value 0.230), the timing of eculizumab after the presentation (*p* value 0.252), the GFR (*p* value 0.43), the protein/creatinine ratio (*p* value 0.700), or the C3 level at the time of discontinuation (*p* value 0.678) (Table 3).

There were no significant differences in the rates of relapse according to the types of gene mutations (*p* > 0.05) (Table 4).

Additionally, there were no significant differences in the relapse times between males and females (mean relapse time for males was 7.84 months, compared to 8.50 months for females). The overall mean time to relapse was 8.04 months. According to the log-rank, it was revealed that X^2^ = 0.143, *p* = 0.705 (Figure 2).

### 3.4. Outcomes

Following discontinuation, all patients who relapsed achieved remission after resuming eculizumab, and none of them progressed to end-stage renal disease (ESRD) (Table 5). 

Five patients showed HUS recurrence episodes once, and 2 patients were subjected to another trial of discontinuation after the relapse episodes were treated. Regarding the renal outcomes, 23 patients were in remission with normal renal function (including 12 (52%) patients who underwent a positive genetic study, and 10 (43.7%) patients with no detected gene mutation), 4 patients had chronic kidney disease (CKD) (three patients had pre-existing chronic kidney disease (CKD) before discontinuation, and one had a DKGE mutation and developed a new CKD after discontinuation), while one patient underwent transplantation

## 4. Discussion

Our study describes one of the largest pediatric aHUS populations. The majority of previously published aHUS studies included both adult and pediatric patients and focused mainly on eculizumab therapy rather than its discontinuation with the consideration of its association with gene mutations at the same time. Eculizumab is a well-tolerated and effective treatment for patients with aHUS [12,13,14]. It has been reported that eculizumab discontinuation is possible and safe in children with aHUS [10]. Several factors reported in the literature should be considered, as they might be associated with the risk of disease progression after discontinuation, such as the age at first presentation, previous TMA manifestations, the prevalence of extrarenal manifestations, and the post-transplantation setting [13]. We did not find in our study an association between the previous factors and risks for disease progression after eculizumab discontinuation. 

The reasons for treatment discontinuation include both medical and economical concerns as well as patient requests [13]. In our study, the most common reason for the discontinuation of eculizumab therapy was that patients experienced remission and their conditions were stable or they were negative for genetic mutations. 

Sixteen (57.2%) patients in this study showed positive genetic testing results, possibly reflected in the high consanguinity percentage (71.4%). The risk of aHUS relapse following eculizumab discontinuation is mainly determined by the presence or absence of gene mutations [10]. Similarly, in our study, seven (87.5%) of the 8 relapsed patients had a detected gene mutation, while patients with no relapse were more likely to have no genetic mutation. Some reports suggest a potential relationship between the type of complement mutation identified and the risk of subsequent clinical manifestations of HUS [13]. Patients with CFH and MCP/CD46 variant mutations appear to be at higher risk of recurrence [1,13,14,15,16,17]. Patients with CFH or THBD/CD141 mutations had the earliest onset of aHUS and the highest mortality rates, whereas patients with MCP/CD46 mutations were associated with the least severe outcomes [16], but there was no significant relationship that could be identified between the type of mutation and the risk of HUS relapse in our study, which may be related to the differences in the background genetics of our population. 

However, the time to recurrence was longer in this study (median of 3 months) than in the study by Ardissino et al. [1] (<6 weeks) and equal to that in the study by Macia et al. [13], but it was shorter than that in the study by Ariceta et al. [15] (median of 5 months). The current literature reports that the rates of HUS recurrence after eculizumab discontinuation are between 22% and 31% [1,10,13,15,18], while in our study, eight patients (28.5%) experienced HUS relapse after discontinuation, making the identification of at-risk patients essential. Contrary to other studies, where patients experiencing HUS recurrence had a shorter duration of treatment [10,13,18], the durations of eculizumab therapy in our study were similar between patients who did and did not suffer HUS recurrence. 

When HUS occurs after discontinuation, restarting eculizumab can prevent further HUS manifestations. Vilalta et al. [17] reported HUS manifesting eight weeks after missing a single eculizumab dose. The rates of eculizumab discontinuation and re-initiation have also been reported for 296 patients receiving eculizumab in the global aHUS Registry. In patients aged <18 years, 28 (24%) discontinued treatment, of whom 7 (25%) restarted eculizumab treatment [19]. Similarly, in our study, of all eight patients who relapsed, eculizumab was restarted within 24–48 h in the majority of them. Most of the patients who relapsed achieved remission after resuming eculizumab. However, it has been reported that the restart of eculizumab could not prevent the deterioration of renal function, HUS recurrence, or subsequent ESKD [14]. We observed that 5 patients had a relapse within 1–3 months, with upper respiratory tract infection being the most common triggering factor for a new recurrence. It remains a debate between researchers about the safety and feasibility of the discontinuation of eculizumab therapy [17,20]. In a selected group of patients, a trial of gradual discontinuation by lengthening the dosing interval of eculizumab seems to be a reasonable and safe alternative to the indefinite continuation of the drug [15,18]. In this study, eleven of our patients underwent eculizumab dose spacing prior to complete discontinuation. We emphasize the importance of the previously suggested recommendation by other studies that in the case of discontinuation, patient education regarding early HUS symptoms should be ensured, along with regular urine dipsticks to monitor hematuria and proteinuria. Additionally, if possible, regular blood tests to monitor hemoglobin, platelet, serum creatinine, and LDH levels are crucial. If HUS recurrence is suspected, eculizumab should be rapidly reinitiated [1,11,21,22]. 

The outcome after discontinuation in this study was fairly good, with the majority of children remaining in remission with normal kidney function, which is better than a previous report by Ariceta et al. [17], as 8.0% of their patients progressed to ESKD. This study had limitations that should be noted, including the retrospective nature of the study and the varying observation intervals. However, we have reported on the whole pediatric aHUS population in the Arabian Gulf region spanning over 9 years, and to the best of our knowledge this is one of the largest studies in a pediatric population.

## 5. Conclusions

Our study supports the possibility of discontinuing eculizumab treatment in children with aHUS, but this decision may depend on identified gene mutations. The majority of children in our cohort remained in remission after the discontinuation of eculizumab.

A full assessment of the risk factors before the decision to discontinue eculizumab is essential, with parents being involved in the decision. Resuming eculizumab immediately after relapse is crucial and effective in most patients. In this study, although relapses happened, resuming eculizumab therapy was effective in controlling the disease.

## Figures and Tables

**Figure 1 children-09-01734-f001:**
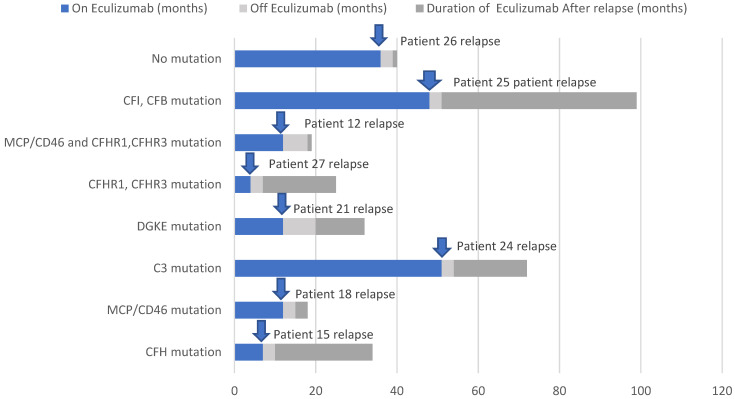
Durations of eculizumab (months) before and after relapse according to the type of gene mutation.

**Figure 2 children-09-01734-f002:**
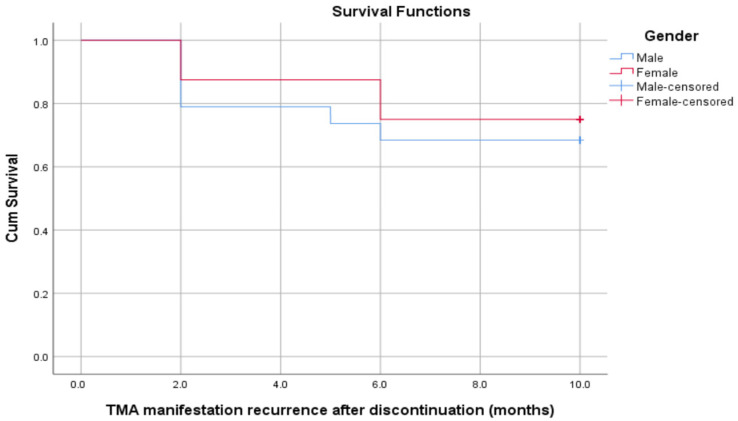
Survival plot for the times to relapse between males and females.

**Table 1 children-09-01734-t001:** Summary of the baseline general characteristics of the patients and laboratory tests at the first presentation of the patients.

Pt #	Sex	Age at Onset	Familial Consanguinity	Family history of aHUS	Hemoglobin on Presentation g/dL	Platelets on Presentation 10^9^/L	LDH on Presentation (U/L)	Creatinine on Presentation µm/L	GFR mL/min/1.73 m^2^ on Presentation	C3 Level on Presentation	Hypertensive	Timing of Eculizumab (After Presentation)	Acute Dialysis on Presentation	Duration While on Eculizumab (Months) before Discontinuation
1	M	54 m	No	Negative	7.00	86,000	3539	174	45	Normal	No	1 day	Yes	2 27
2	F	6 m	No	Sibling	7.60	86,000	3616	278	35	Normal	Yes	2 days	Yes	3
3	M	6 y	Yes	Cousin	10.00	73,000	2008	191	40	Low	Yes	1 day	No	3
4	M	6 y	Yes	Cousin	8.90	50,000	3370	282	20	Low	Yes	1 day	Yes	6
5	M	4 y	Yes	Cousin	7.80	35,000	2182	169	25	Low	No	2 days	Yes	2
6	F	6 y	Yes	Negative	10.00	71,000	1780	38	100	Normal	No	1 day	No	10
7	F	20 m	Yes	Negative	6.20	46,000	2621	441	20	Normal	No	1 day	No	26
8	M	7 y	Yes	Negative	8.00	41,000	2991	207	35	Normal	Yes	30 days	Yes	12
9	F	2 y	Yes	Negative	7.80	77,000	2336	275	20	Normal	No	2 days	Yes	6
10	M	14 m	No	Negative	5.00	11,000	3531	137	22	Low	No	1 day	Yes	19
11	M	9 m	Yes	Negative	8.10	167,000	551	97	95	Normal	Yes	NA	No	2
12	F	3 y	No	Negative	10.60	44,000	2242	164	27	Low	No	3 days	No	12
13	M	3 m	Yes	Cousin	5.20	53,000	1840	300	18	Normal	Yes	9 years	Yes	15
14	F	3 m	Yes	Cousin	8.60	92,000	1760	300	19	Normal	Yes	5 years	Yes	15
15	M	10 m	Yes	Negative	7.50	200,000	400	252	20	Normal	Yes	5 days	No	7
16	M	5 y	Yes	Negative	12.00	282,000	379	160	25	Low	Yes	3 months	Yes	12
17		5 y	No	Negative	9.10	96,000	391	203	28	Low	Yes	7 days	No	11
18	M	7 y	Yes	Negative	8.20	13,000	4020	195	24	Low	Yes	7 days	Yes	12
19	M	2 y	No	Negative	8.70	6000	4848	393	20	Normal	No	5 days	Yes	5
20	F	4 y	Yes	Negative	9.30	121,000	631	291	16	Low	No	1 week	No	4
21	M	1 m	Yes	Sibling	6.20	32,000	1118	709	12	Low	Yes	7 days	No	12
22	M	9 y	Yes	Negative	5.80	81,000	776	511	16	Low	No	3 days	Yes	20
23	M	11 y	No	Sibling	12.10	59,000	3274	504	17	Low	Yes	72 h	Yes	48
24	M	9 m	Yes	Sibling	7.50	47,000	1023	80	16	Normal	Yes	7 days	Yes	51
25	M	20 m	Yes	Cousin	7.70	321,000	433	165	24	Low	Yes	4 months	No	48
26	M	10 m	Yes	Sibling	6.10	70,000	519	101	29	Low	Yes	4 days	Yes	36
27	F	4 y	No	Negative	8.60	23,000	1896	182	20	Low	Yes	4 days	Yes	4
28	F	2 m	Yes	Negative	7.70	104,000	4904	80	23	Normal	No	59 days	Yes	5

M: male; F: female.

**Table 2 children-09-01734-t002:** Genetic variants of the patients.

Pt #	Genetic Testing Result	Variant Nomenclature	Zygosity	Variant Classification	Relapse	Outcome
1	No mutation	NA	NA	NA	No	In remission
2	No mutation	NA	NA	NA	No	In remission
3	MCP/CD46 CFHR1, CFHR3 mutation	c.736T > A; p.Phe246Ile Whole genes deletion	Homozygous Homozygous	VOUS VOUS	No	In remission
4	CFHR1, CFHR3 mutation	Whole genes deletion	Homozygous	VOUS	No	In remission
5	No mutation	NA	NA	NA	No	In remission
6	No mutation	NA	NA	NA	No	In remission
7	No mutation	NA	NA	NA	No	In remission
8	No mutation	NA	NA	NA	No	In remission
9	Not done	NA	NA	NA	No	In remission
10	CFH mutation	c.2195C > T; p.Thr732Met	Heterozygous	VOUS	No	In remission
11	CFHR1, CFHR3 mutation	Whole genes deletion	Homozygous	VOUS	No	In remission
12	CD46 CFHR1-CFHR3 mutation	c.350_351dup;p.Glu118Thrfs17 Whole genes deletion	Homozygous Homozygous	Likely pathogenic VOUS	Yes	In remission
13	DGKE mutation				No	CKD
14	DGKE mutation				No	CKD
15	CFH mutation	c.3545G > T; p.Arg118Met	Heterozygous	VOUS	Yes	Transplant
16	CFHR1, CFHR3 mutation	Whole genes deletion	Heterozygous	VOUS	No	In remission
17	No mutation	NA	NA	NA	No	CKD
18	MCP/CD46 mutation	c.608T > C; p.Ile203Thr	Homozygous	VOUS	Yes	In remission
19	CFHR1, CFHR3 mutation	Whole genes deletion	Homozygous	VOUS	No	In remission
20	CFHR1, CFHR3 mutation	Whole genes deletion	Heterozygous	VOUS	No	In remission
21	DGKE	c.413 G > A (p.Cys138Tyr)	Homozygous	Likely pathogenic	Yes	CKD
22	No mutation	NA	NA	NA	No	In remission
23	MCP/CD46 CFI	c.736T > A (p.Phe246Ile) c.540 A > G (p.Glu180Glu)	Homozygous Heterozygous	VOUS VOUS	No	In remission
24	C3 mutation	c.3326 T > G (P.leu1109 Arg)	Homozygous	Likely pathogenic	Yes	In remission
25	C3 CFB CFI	c.3326T > G (p. Leu1109Arg) c.1697A > C (p. Glu566Ala) c.1246A > C (p. Ile416 Leu)	Heterozygous Heterozygous Homozygous	VOUS Likely pathogenic Pathogenic	Yes	In remission
26	No mutation	NA	NA	NA	Yes	In remission
27	CFHR1, CFHR3 mutation	Whole genes deletion	Homozygous	VOUS	Yes	In remission
28	CFHR1, CFHR3 mutation	Whole genes deletion	Homozygous	VOUS	No	In remission

CFH: complement factor H; CFB: complement factor B; CFI: complement factor I; MCP: membrane cofactor protein; CFHR1, CFHR3: complement factor H-related 1, -3; VOUS: variant of uncertain significance; NA: not available; DGKE: diacylglycerol kinase epsilon; CKD: chronic kidney disease.

**Table 3 children-09-01734-t003:** Characteristics of patients who relapsed and patients who did not relapse before eculizumab discontinuation.

Factor	With Relapse	No Relapse	*p*-Value
	N (%)	N (%)	
	(*n* = 8)	(*n* = 20)	
Age at onset HUS			
<1 year	04 (50.0%)	04 (20.0%)	0.505
1–3 years	02 (25.0%)	04 (20.0%)	
4–5 years	01 (12.5%)	05 (25.0%)	
6–8 years	01 (12.5%)	03 (15.0%)	
9–12 years	0	04 (20.0%)	
Creatinine at presentation, micromole/L (mean ± SD)	231.0 ± 200.4	251.6 ± 133.1	0.753
C3 Level at the presentation			
Low	05 (62.5%)	09 (45.0%)	0.678
Normal	03 (37.5%)	11 (55.0%)	
Timing of eculizumab after the presentation			
<2 days	0	07 (35.0%)	0.252
2–4 days	03 (37.5%)	04 (20.0%)	
>4 days	04 (50.0%)	06 (30.0%)	
Unknown	01 (12.5%)	03 (15.0%)	
Acute dialysis before eculizumab			
Yes	06 (75.0%)	12 (60.0%)	0.669
No	02 (25.0%)	08 (40.0%)	
Duration while on eculizumab (months)			
1–3	0	04 (20.0%)	0.23
>3–6	01 (12.5%)	05 (25.0%)	
>6–12	02 (25.0%)	02 (10.0%)	
>12–24	01 (12.5%)	06 (30.0%)	
>24–48	03 (37.5%)	3 (5.0%)	
>48–60	01 (12.5%)	0	
Complement gene variants			
CFH mutation	01 (12.5%)	01 (05.0%)	0.497
MCP/CD46 mutation	01 (12.5%)	01 (05.0%)	0.497
C3 mutation	01 (12.5%)	0	0.286
DGKE mutation	01 (12.5%)	02 (10.0%)	1.000
CFHR1, CFHR3 mutation	01 (12.5%)	04 (20.0%)	1.000
MCP/CD46 and CFHR1, CFHR3 mutation	01 (12.5%)	01 (05.0%)	0.497
CFI, CFB mutation	01 (12.5%)	0	0.286
No mutation	01 (12.5%)	10 (50.0%)	0.088
Creatinine at time discontinuation of eculizumab	63.7 ± 76.6	43.2 ± 19.9	0.267
GFR at time of discontinuation of eculizumab (mean ± SD)	101.9 ± 32.1	109.1 ± 16.5	0.700
Protein/creatinine ratio (g/mmol) at time of discontinuation of eculizumab (median (min–max)) (mean ± SD)	0.45 (0.01–1.90)	0.50 (0.01–15)	0.439
Durations of eculizumab treatment in months, median (min–max)	12.0 (4.0–51.0)	10.5 (2.0–48.0)	0.165
GFR (mL/min/1.73 m^2^) at the last visit after discontinuation			
≥90	06 (75.0%)	16 (80.0%)	
60–89	01 (12.5%)	02 (10.0%)	1000
30–59	01 (12.5%)	02 (10.0%)	

**Table 4 children-09-01734-t004:** Rates of relapse according to gene mutations.

Gene Mutations	Rate of Relapse			*p*-Value *
	One N (%)	Two N (%)	Three N (%)	
CFH mutation	01 (20.0%)	0	0	1.000
MCP/CD46 mutation	0	0	01 (100%)	0.125
C3 mutation	01 (20.0%)	0	0	1.000
DGKE mutation	0	01 (50.0%)	0	0.375
CFHR1, CFHR3 mutation	01 (20.0%)	0	0	1.000
MCP/CD46 and CFHR1, CFHR3 mutation	01 (20.0%)	0	0	1.000
CFI, CFB mutation	0	01 (50.0%)	0	0.375
No mutation	01 (20.0%)	0	0	1.000

Note: * *p*-value calculated using Fischer’s exact test.

**Table 5 children-09-01734-t005:** The individual characteristics and outcomes of 8 patients who experienced aHUS relapse.

Patient	Age at Initial Eculizumab Treatment (Years)	Mutation	Reason for Discontinuation Time	Time of Eculizumab before Discontinuation	Time of Relapse	Restarted Eculizumab	Outcome	Duration of Follow Up Months
Patient 12	6 years	MCP/CD46, CFHR1, CFHR3 mutation	Family preference	12 months	3 months	Yes	Stable (in remission)	15
Patient 15	6 years	CFH mutation	Stable patient and complete remission	7 months	3 months	Yes	Transplanted	16
Patient 18	7 years	MCP/CD46 mutation	Stable patient and complete remission	12 months	15 months	Yes	Stable (in remission)	48
Patient 21	14 months	DGKE mutation	DKGE mutation	12 months	12 months	Yes	CKD	60
Patient 24	3 months	C3 mutation	Genetic testing showed DKGE mutation	51 months	3 months	Yes	Stable (in remission)	30
Patient 25	3 months	CFI, CFB mutation	Genetic testing showed DKGE mutation	48 months	3 months	Yes	Stable (in remission)	17
Patient 26	2 years	No mutation	Insurance coverage	36 months	7 weeks	Yes	Stable (in remission)	24
Patient 27	9 months	CFHR1, CFHR3 mutation	Stable patient and complete remission	4 months	14 months	Yes	Stable (in remission)	60

## Data Availability

The datasets generated and analyzed during the current study are available from the corresponding author upon reasonable request.

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
