# Peer review of "A Multicenter Study Evaluating the Discontinuation of Eculizumab Therapy in Children with Atypical Hemolytic Uremic Syndrome"

_children, 2022, doi:10.3390/children9111734_

Round 1

Reviewer 1 Report

The authors of this manuscript describe potential risk factors of aHUS relapse when approved pharmacologic therapy (Eculizumab) was discontinued, reasons for treatment discontinuation, and patient-specific outcomes.

Comments:

1) The authors state that there was no significant relationship between relapse and the GFR (lines 179-181) and refer to Table 3. However, Table 3 only lists the creatinine at time of ECZ discontinuation. To be clear and consistent, the authors should list the GFR (mL/min/1.73 m2) in the relapse group and no-relapse group in this Table 3, to reflect their statement.

2) It would also be helpful to know eGFR at presentation in Table 1, to compare to endpoint eGFR at time of ECZ discontinuation in Table 3.

3) The authors nicely provide patient-specific outcomes in Table 5 (remission vs CKD vs transplanted). It would be helpful to know each subject's duration of follow up leading to the reported outcome in Table 5.

4) The authors use CKD as a reported outcome. Please define what you are using as a CKD definition (ie eGFR <60 - those with CKD 3 or higher? or eGFR <90 - those with CKD 2 or higher?).

5) With some patients having positive pathogenic mutations, this may be why many of the patients seemed to present young. But, did age at disease onset factor into risk of disease relapse in the analysis?

6) Table 1 was difficult to read as many of the lab values and words wrapped into the subsequent lines. Please reformat this.

Author Response

good morning 

hope you are doing fine 

  • The authors state that there was no significant relationship between relapse and the GFR (lines 179-181) and refer to Table 3. However, Table 3 only lists the creatinine at time of ECZ discontinuation. To be clear and consistent, the authors should list the GFR (mL/min/1.73 m2) in the relapse group and no-relapse group in this Table 3, to reflect their statement.

 I added the GFR in table 3

  • It would also be helpful to know eGFR at presentation in Table 1, to compare to endpoint eGFR at time of ECZ discontinuation in Table 3.

I added the eGFR at presentation in table 1

  • The authors nicely provide patient-specific outcomes in Table 5 (remission vs CKD vs transplanted). It would be helpful to know each subject's duration of follow up leading to the reported outcome in Table 5

It has been added to table 5

  • The authors use CKD as a reported outcome. Please define what you are using as a CKD definition (ie eGFR <60 - those with CKD 3 or higher? or eGFR <90 - those with CKD 2 or higher?)

KDIGO guidline was  used , it has been added to the text line 111,112

  • With some patients having positive pathogenic mutations, this may be why many of the patients seemed to present young. But, did age at disease onset factor into risk of disease relapse in the analysis?

I added the age at disease onset in Table 3

  • Table 1 was difficult to read as many of the lab values and words wrapped into the subsequent lines. Please reformat this.

Done

Reviewer 2 Report

Lines 37-40 “The risk factors of recurrence were positive genetic mutations (p=0.020). On the other hand, there was no significant relationship between the relapse and the need for acute dialysis, the duration of eculizumab therapy before discontinuation, and duration of eculizumab withdrawal before relapse.”

These statements are not clearly supported by the data presented in the text.

Lines 41-44 “…,while 4 patients had chronic kidney disease 41 (CKD) including three patients had pre-existing chronic kidney disease (CKD) before discontinuation, and one with DKGE mutation developed a new CKD after discontinuation, one patient underwent transplantation.”

This phrase is ambiguous and unclear of what it intends to mean. Are there 4 or 5 cases with deteriorated kidney function after discontinuation of eculizumab?  It should be clarified.

Results

Lines 129-130 “The mean values of hemoglobin, platelets, and LDH were 8.1, 84.5, and 2106, respectively.”

Clarify at what time points of the courses these values were obtained.

The reference ranges of all laboratory tests should be provided.

Line 166 Figure 1

The durations of eculizumab treatment and treatment discontinuation before relapse should be depicted for all cases, not just relapse cases. For those without relapse, the ongoing duration of treatment discontinuation should be depicted and statistically compared with those with relapse.

Relapse of aHUS

Lines 178-179 “The prevalence of no relapse was significantly more common among those with no mutation (p=0.020).”

This statement is problematic.

Firstly, how do the authors arrive at the stated P-value? A table to depict the analysis that supports this P-value should be included for clarity. Based on the data presented, there was one case of relapse among 11 cases without any detectable variant and 7 cases of relapse among 16 cases with at least one detectable variant. If this is correct, the P-value cannot be 0.02 per Fisher’s test.

Secondly, DGKE nephropathy has a different pathophysiology. The role of eculizumab is questionable, except in occasional cases that may have concurrent complement dysregulation. For the objective of this study, DGKE cases should not be included. Similarly, CFHR1-CFHR3 deletion is a common allele in some populations. Its mere detection should not be equated with a role in the pathogenesis of complement dysregulation, unless anti-CFH antibody was detected.

Overall, these two groups of cases should be analyzed separated.

Line 183 Table 3 depicts results that are not quite relevant for the main objective of this study. It should revised to depict the renal status (serum creatinine, eGFR, proteinuria and hematuria), blood pressure and relevant symptoms at the time of treatment discontinuation and their optima conditions achieved after the relapse is over. Since renal injury and blood pressure elevation may occur sub-clinically and symptoms of mild relapse may be quite mild or non-specific, these parameters should also be depicted and analyzed for the cases without “relapse”, at the time of treatment discontinuation and also at the latest follow-up.

Line 194 Figure 2 is irrelevant and should be deleted.

Line 200 Table 5 should be deleted. Its relevant data should be incorporated in the recommended Table 3.

Line 211  The section of Discussion, comprising a single lengthy paragraph, is difficult to follow. It should be divided into paragraphs, with each focusing on a specific subject of relevance. The discussion should be more focused and based on the results presented. For example, the risk of relapse for those without a detectible variant, 0.091 (1/11), has a very broad 95% confidence interval (0.002-0.41). Therefore, the small numbers of cases preclude an exact assessment of relapse risk.

The concerns of sub-clinical disease activity leading to subtle organ dysfunctions, elevated blood pressures and/or progressive renal failure in those receiving maintenance prophylactic treatment should be addressed.

Tables: The type-settings of the tables are quite poor and need improvement.

Author Response

good morning 

hope you  are doing fine 

Lines 37-40 “The risk factors of recurrence were positive genetic mutations (p=0.020). On the other hand, there was no significant relationship between the relapse and the need for acute dialysis, the duration of eculizumab therapy before discontinuation, and duration of eculizumab withdrawal before relapse.”

These statements are not clearly supported by the data presented in the text.

It has been to the text line 188- 192

Lines 41-44 “…,while 4 patients had chronic kidney disease 41 (CKD) including three patients had pre-existing chronic kidney disease (CKD) before discontinuation, and one with DKGE mutation developed a new CKD after discontinuation, one patient underwent transplantation.”

This phrase is ambiguous and unclear of what it intends to mean. Are there 4 or 5 cases with deteriorated kidney function after discontinuation of eculizumab?  It should be clarified

It has been clarified in line 40-44

Results

Lines 129-130 “The mean values of hemoglobin, platelets, and LDH were 8.1, 84.5, and 2106, respectively.”

Clarify at what time points of the courses these values were obtained.

The reference ranges of all laboratory tests should be provided.

It has been clarified in line 132 -133

Line 166 Figure 1

The durations of eculizumab treatment and treatment discontinuation before relapse should be depicted for all cases, not just relapse cases. For those without relapse, the ongoing duration of treatment discontinuation should be depicted and statistically compared with those with relapse

It has been clarified in line 132 -133

Relapse of aHUS

Lines 178-179 “The prevalence of no relapse was significantly more common among those with no mutation (p=0.020).”

This statement is problematic.

Firstly, how do the authors arrive at the stated P-value? A table to depict the analysis that supports this P-value should be included for clarity. Based on the data presented, there was one case of relapse among 11 cases without any detectable variant and 7 cases of relapse among 16 cases with at least one detectable variant. If this is correct, the P-value cannot be 0.02 per Fisher’s test.

P value was not per Fisher’s test, but per Chi- sequare test

Secondly, DGKE nephropathy has a different pathophysiology. The role of eculizumab is questionable, except in occasional cases that may have concurrent complement dysregulation. For the objective of this study, DGKE cases should not be included. Similarly, CFHR1-CFHR3 deletion is a common allele in some populations. Its mere detection should not be equated with a role in the pathogenesis of complement dysregulation, unless anti-CFH antibody was detected.

Overall, these two groups of cases should be analyzed separated.

DGKE nephropathy   has different pathophysiology  but it is associated with HUS . it was included also  in other studies like Eculizumab discontinuation in atypical haemolytic uraemic syndrome: TMA recurrence risk and renal outcomes.

Ariceta G, Fakhouri F, Sartz L, Miller B, Nikolaou V, Cohen D, et al. Eculizumab discontinuation in atypical haemolytic uraemic syndrome: TMA recurrence risk and renal outcomes. Clin Kidney J. 2021;(January):1–10.

  • All CFHR1-CFHR3 mutation in this study were pathogenic,   we did nit includ any heterozygous mutation

Line 183 Table 3 depicts results that are not quite relevant for the main objective of this study. It should revised to depict the renal status (serum creatinine, eGFR, proteinuria and hematuria), blood pressure and relevant symptoms at the time of treatment discontinuation and their optima conditions achieved after the relapse is over. Since renal injury and blood pressure elevation may occur sub-clinically and symptoms of mild relapse may be quite mild or non-specific, these parameters should also be depicted and analyzed for the cases without “relapse”,

at the time of treatment discontinuation and also at the latest follow-up

We included serum creatinine, eGFR, proteinuria, BP

Line 194 Figure 2 is irrelevant and should be deleted.

we think that figure 2 better to keep it  includes Duration of Eculizumab (months) before and after relapse according to the type of gene mutation.

Line 200 Table 5 should be deleted. Its relevant data should be incorporated in the recommended Table 3.

we think that table  5  better to keep it, it included relapsing HUS and their characteristics

Line 211  The section of Discussion, comprising a single lengthy paragraph, is difficult to follow. It should be divided into paragraphs, with each focusing on a specific subject of relevance. The discussion should be more focused and based on the results presented. For example, the risk of relapse for those without a detectible variant, 0.091 (1/11), has a very broad 95% confidence interval (0.002-0.41). Therefore, the small numbers of cases preclude an exact assessment of relapse risk.

Done

The concerns of sub-clinical disease activity leading to subtle organ dysfunctions, elevated blood pressures and/or progressive renal failure in those receiving maintenance prophylactic treatment should be addressed.

Done

Tables: The type-settings of the tables are quite poor and need improvement.

Done
